# Influence of Neutron Irradiation on Microstructure and Mechanical Properties of Coarse- and Ultrafine-Grained Titanium Grade 2

Pavel Zháňal [1], Tomáš Krajňák [2,3], Mariia Zimina [1,4], Alica Fedoriková [1], Ondřej Srba [1], Petr Harcuba [2], Josef Stráský [2] and Miloš Janeček [2,*]

1  Research Centre Řež, Material Analysis, 250 68 Husinec-Řež, Czech Republic
2  Faculty of Mathematics and Physics, Charles University, 121 16 Praha, Czech Republic
3  Research Centre, University of Žilina, 01026 Žilina, Slovakia
4  Interface Analysis Centre, School of Physics, University of Bristol, Bristol BS8 1FD, UK
*  Correspondence: janecek@met.mff.cuni.cz; Tel.: +420-95155-1358

**Abstract:** The influence of neutron irradiation on the microstructure and related mechanical properties of Ti Grade 2 in coarse- and ultrafine-grained conditions was investigated. It was found that mechanical properties of the coarse-grained (CG) state were significantly affected by neutron irradiation. At room temperature (RT), the yield stress increased by more than 30%, whereas the ductility decreased by more than 50%. An even bigger difference in the mechanical properties between irradiated and non-irradiated states was observed at a temperature of 300 °C. Changes in the mechanical properties can be attributed to the high density of defect clusters/dislocation loops induced by neutron irradiation. On the other hand, the ultrafine-grained (UFG) state is more resistant to radiation damage. The mechanical properties at RT did not change upon neutron radiation, while at a temperature of 300 °C, the yield stress increased only by about 10%. Enhanced radiation resistance of the UFG state can be attributed to the presence of a high density of dislocations and dense network of high-angle grain boundaries, which act as traps for radiation-induced defects and, thus, prevent the accumulation of these defects in the microstructure.

**Keywords:** neutron irradiation; grain size; defect clusters; dislocation loops; yield stress

## 1. Introduction

The development of small modular nuclear reactors as well as prolonged lifetime of existing nuclear power plants call for new materials utilizable in the harsh environment of nuclear reactors. Materials with improved functional properties, such as mechanical properties at room and elevated temperatures, corrosion resistance and radiation resistance, are demanded. The successful implementation of these requirements depends mainly on the application of metals and alloys characterized by high radiation resistance and an accelerated decrease in induced radioactivty [1–3]. It was shown in [4] that titanium alloys T1-70A and T1-6A1-4V are highly radiation resistant. Void swelling, which is the dominant mechanism of radiation damage in 316SS and other candidate materials, is either nonexistent or negligible in titanium alloys at temperatures below 500 °C. Moreover, Ti-70A and Ti-6A1-4V were more resistant to helium embrittlement than 316SS. A titanium component will cause only minor radiological hazard 8 years after reactor shutdown and will, therefore, reduce the waste disposal costs/concerns from reactor systems. In comparison, a 316SS component will be 23,000 × more radioactive after 8 years [2]. Titanium alloys, despite their higher manufacturing cost, have the potential to improve the safety and lifetime of nuclear reactors and, subsequently, reduce the maintenance and disposal costs of structures utilized in nuclear power plants. Mechanical properties of metallic materials change due to the introduction of lattice defects. The formation of lattice defects during irradiation is regarded

as an adverse process as it potentially leads to embrittlement and failure. On the other hand, the introduction of lattice defects during processing of materials by plastic deformation is intentional and aims to increase the strength. The most extreme processes of plastic deformation are so-called severe plastic deformation (SPD) methods [5]. SPD processing induces high dislocation density and ultimately results in significant grain refinement. Ultrafine-grained (UFG) materials are characterized by grains smaller than 1 μm separated by high-angle grain boundaries [6]. The general idea of utilizing UFG materials as radiation-tolerant materials is that the dense network of grain boundaries serves as sinks for radiation-induced lattice defects, such as vacancies, dislocations or interstitials. In other words, during irradiation, the rates of defect formation and annihilation in the UFG structure are balanced, resulting in unchanging mechanical properties. Indeed, several works studying the impact of irradiation on properties of UFG materials delivered promising results, showing that these materials are very resistant to radiation damage and have a generally excellent balance of properties, including outstanding fatigue resistance [7–11]. It must be noted that mechanical properties of UFG Ti do not change significantly after exposure to 450 °C [12,13]; thermal stability of UFG Ti, therefore, seems to be sufficient for utilization in standard water-cooled reactors with working temperatures well below 400 °C. The aim of this study is to investigate the influence of neutron irradiation on the mechanical properties of UFG Ti Grade 2 produced by the SPD method of ECAP-Conform.

## 2. Materials and Methods

Extruded commercial Ti Grade 2 was used as the benchmark material (CG), whereas UFG material was prepared via the ECAP-Conform technique. The tensile samples were neutron irradiated in the reactor LVR-15 at the Research Center Řež, Czech Republic. After 6 months, the final dose was 0.3 dpa (displacement per atom) and samples exhibited activity of $1.95 \times 10^{-6}$ Bq. Tensile tests were performed at room (RT) and elevated temperature (300 °C) and a constant strain rate of $10^{-3}$ s$^{-1}$ using a servo-hydraulic machine Instron 8874 in SUSEN Hot Cells (Instron, Notwood, MA, USA). The samples for tensile testing were machined according to ISO 6892-1 with a gauge length of 15.75/10.5 mm and diameter of 3/2 mm for the CG and UFG state, respectively. Three independent tests were performed for each condition. Transmission electron microscopy (TEM) was performed using JEOL 2200FS (Jeol Ltd., Tokyo, Japan) at 200 kV. The samples were cut from the head of tested tensile samples. TEM samples were ground to a thickness of 0.2 mm and electrochemically polished using an electrolyte containing 6% $HClO_4$, 33% $C_4H_{10}O$ (butanol) and 61% $CH_3OH$ (methanol). TEM specimens were further used for SEM and EBSD analysis (Tescan, Mira3) (Ametek, Inc. Berwyn, PA, USA) to determine the grain size.

## 3. Results

### 3.1. Microstructure of Irradiated and Non-Irradiated Ti Grade 2

The microstructure of the CG state is homogenous and consists of equiaxed grains with a size of about 12 μm (Figure 1a). Exposition of the CG sample to a temperature of 300 °C did not affect the grain size (Figure 1b). Similarly, no significant difference in the dislocation structure of both samples was observed as it is apparent from TEM micrographs (Figure 2a,b). On the other hand, the substantial change in the dislocation structure can be observed in the microstructure of the CG sample upon irradiation, see Figure 2c. A high density of homogeneously distributed defect clusters/dislocation loops is visible inside the original grains. Dislocation loops tend to have preferential orientation within the individual grains, and these preferential orientations vary in the neighboring grains. The neutron-induced defect clusters/dislocation loops are preserved in the microstructure during the annealing at 300 °C, as is apparent from Figure 2d. Figure 3 summarizes the TEM micrographs of the UFG samples in both non-irradiated and irradiated conditions. The UFG microstructure of non-irradiated sample consists of grains with a size in the range of hundreds of nanometers (the average grain size as determined by ACOM-TEM was about 100 nm) that contain a high density of dislocations, cf. Figure 3a. In contrast to the

CG state, no additional change in the microstructure of the UFG sample was observed upon the neutron irradiation, see Figure 3c. Microstructure recovery at a temperature of 300 °C can be observed in both the non-irradiated and irradiated UFG states (Figure 3b,d), resulting in a decrease in dislocation density inside the previously deformed grains and in a slight grain growth.

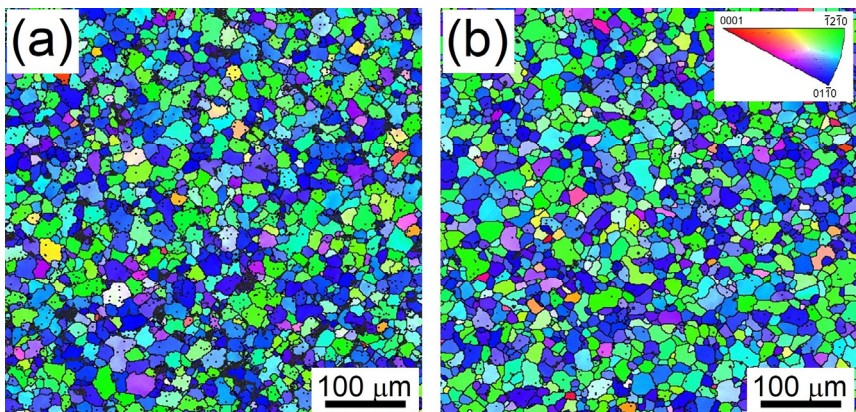

**Figure 1.** EBSD maps of CG Ti Grade 2 at (**a**) RT and (**b**) 300 °C.

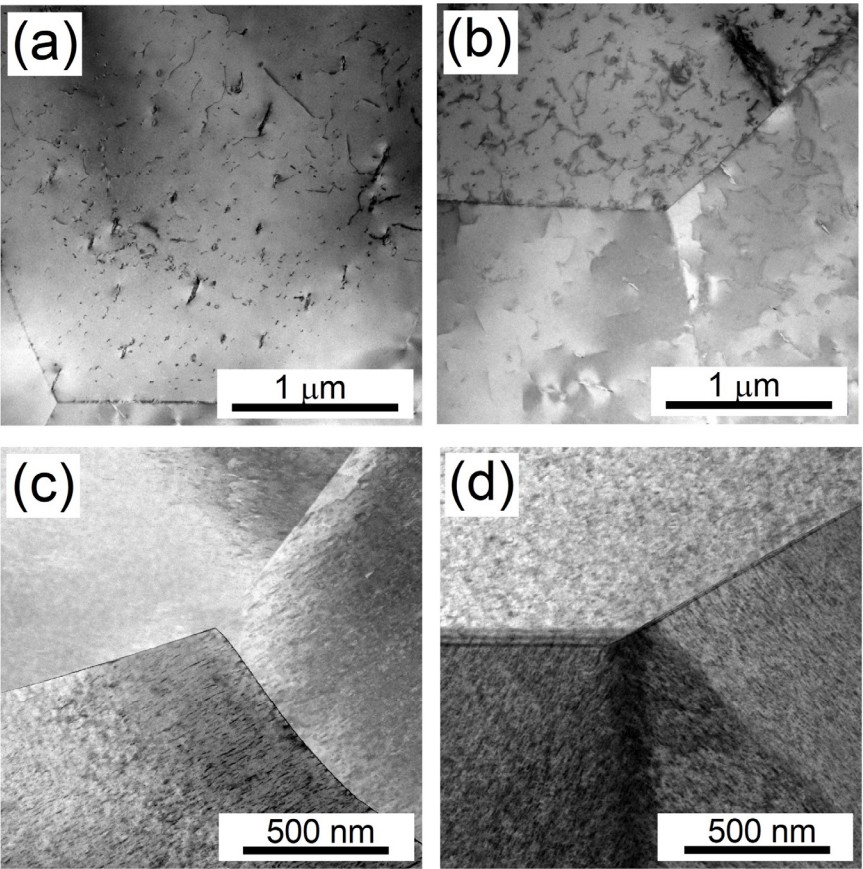

**Figure 2.** TEM micrographs of non-irradiated CG Ti Grade 2 at (**a**) RT, (**b**) 300 °C and irradiated CG Ti Grade 2 at (**c**) RT, (**d**) 300 °C.

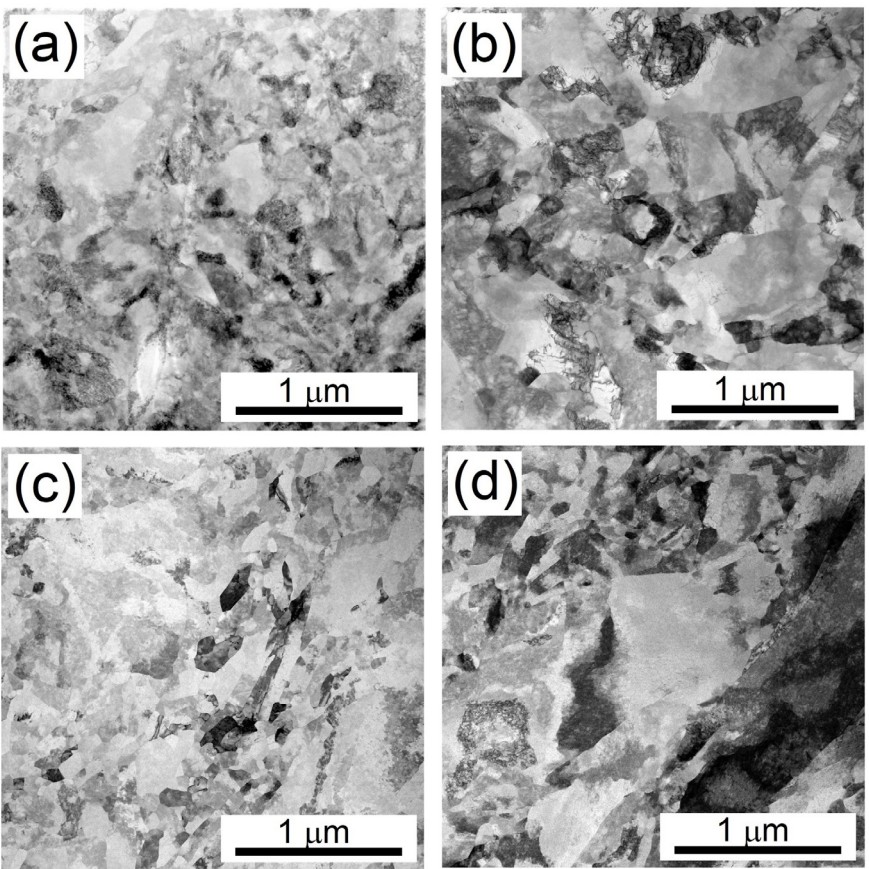

**Figure 3.** TEM micrographs of non-irradiated UFG Ti Grade 2 at (**a**) RT, (**b**) 300 °C and irradiated UFG Ti Grade 2 at (**c**) RT, (**d**) 300 °C.

### 3.2. Mechanical Properties of Irradiated and Non-Irradiated Ti Grade 2

A significant influence of irradiation on the mechanical performance of the CG state at RT was observed (Figure 4a). The yield stress ($\sigma_{0.2}$) of the irradiated CG specimen increased by more than 30%, whereas the ductility decreased by about 50%. At a temperature of 300 °C, the influence of irradiation on mechanical performance is even stronger; the yield stress increased by more than 100% while the ductility decreased by 80%. As expected, the non-irradiated UFG material exhibited significantly higher yield stress and reduced ductility than its CG counterpart at both temperatures. However, almost no influence of irradiation on mechanical performance, in particular on the ductility, was observed in the UFG material, both at RT and 300 °C (Figure 4c,d). The most striking result is that the ductility of irradiated UFG material is similar to the irradiated CG material (at RT) or even higher (at 300 °C). Mechanical properties of investigated samples are summarized in Table 1.

**Table 1.** Mechanical properties of studied samples (non-irradiated (NI) and irradiated (I)).

| Material | Temperature | Condition | $\sigma_{0.2}$ (MPa) | UTS (MPa) |
|---|---|---|---|---|
| CG Ti Grade 2 | RT | NI | 643 ± 20 | 722 ± 22 |
| | | I | 866 ± 26 | 906 ± 27 |
| | 300 °C | NI | 163 ± 10 | 327 ± 10 |
| | | I | 388 ± 12 | 415 ± 13 |
| UFG Ti Grade 2 | RT | NI | 1140 ± 34 | 1202 ± 36 |
| | | I | 1196 ± 36 | 1323 ± 39 |
| | 300 °C | NI | 574 ± 17 | 704 ± 20 |
| | | I | 666 ± 20 | 780 ± 23 |

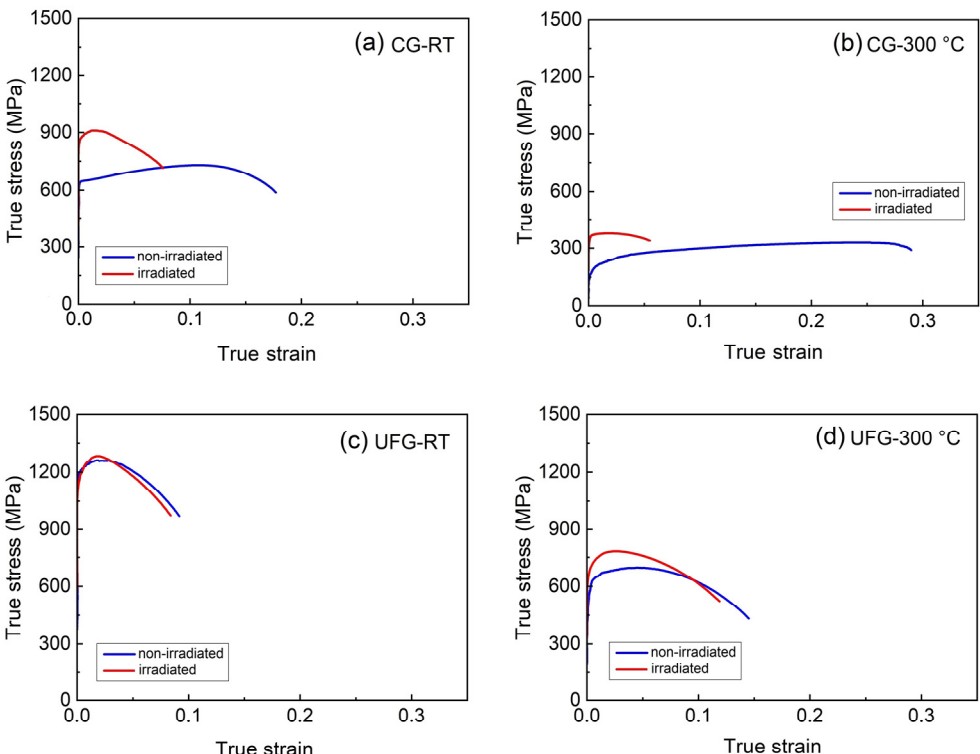

**Figure 4.** True stress–true strain tensile curves of CG Ti Grade 2 at (**a**) RT, (**b**) 300 °C and UFG Ti Grade 2 at (**c**) RT, (**d**) 300 °C.

## 4. Discussion

Detailed TEM investigation indicated that the microstructural response of Ti grade 2 on the neutron irradiation to a dose level of 0.3 dpa depends significantly on the grain size. In the coarse-grained material, a high density of defect clusters/dislocation loops was observed upon irradiation. In addition, the density of these irradiation-induced defects did not decrease after annealing at 300 °C. Similarly, the small defect clusters and coarser dislocations loops were observed in the Ti5Al2.5Sn ($\alpha$) alloy of comparable grain size (about 20 μm) irradiated by neutrons to the same dose level at temperatures of 50 °C and 350 °C, respectively [14]. On the other hand, no remarkable changes were observed in the microstructure of UFG material upon irradiation. It can be attributed to the presence of a dense network of high-angle grain boundaries along with the high dislocation density stored in the original UFG material [15,16]. Radiation-induced point defects (vacancies or interstitials) can be absorbed at grain boundaries because the diffusion of these point defects along grain boundaries (pipe diffusion) is substantially faster than the bulk diffusion [17]. Hence, a vacancy or an interstitial absorbed at the grain boundary is quickly transported to the surface where it disappears. Because of this mechanism, grain boundaries can, in principle, absorb all point defects created by radiation since defects absorbed at grain boundaries are not stored there but are quickly transported to the surface. As a result, the defects induced during irradiation of UFG material are removed by two processes. They can either annihilate with the already present dislocations or can be absorbed by the grain boundaries. The density of dislocations in irradiated UFG specimens remains, therefore, almost unchanged. Due to the above-described microstructural changes, the mechanical response of the Ti grade 2 in the uniaxial tensile testing was found to depend on the grain size. Whereas significant hardening accompanied by the reduced uniform elongation was observed in the CG state upon irradiation at both testing temperatures (RT and 300 °C), mechanical performance of the UFG material remained almost unchanged. Only a slight increase in the yield stress and a reduction in elongation by about 20 and 10% were observed at the testing temperature of 300 °C, respectively. Neutron radiation

hardening was described also in the coarse-grained Ti6Al4V ($\alpha$ + $\beta$) and Ti5Al2.5Sn ($\alpha$) alloys tested in tension at both 50 °C and 350 °C [14]. The hardening of irradiated CG samples can be attributed mainly to the radiation-induced defect clusters/dislocation loops, which act as obstacles for dislocation movement. Dislocation loops in neutron-irradiated materials are formed by the collapse of large vacancy clusters [18]. Energetic neutrons cause atom displacements and introduce Frenkel pairs (vacancies and interstitials). Vacancies in Ti at room temperature are mobile and agglomerate into vacancy clusters. The driving force for the agglomeration is the reduction in surface energy. Large vacancy clusters collapse into dislocation loops, i.e., from a 3D defect into a 2D defect, since dislocation loops have lower energy. The typical development sequence of defects in neutron-irradiated material is vacancies → small vacancy clusters → large vacancy clusters → dislocation loops. Similarly, interstitials agglomerate into clusters, which develop into dislocation loops as well.

In summary, Ti grade 2 with UFG microstructure exhibits significant resistance to neutron radiation. No irradiation-induced precipitation or change in dislocation density was observed. Ti grade 2 in the UFG condition can be, therefore, considered as a promising candidate for utilization in standard water-cooled reactors.

## 5. Conclusions

Neutron radiation resistance of the Ti Grade 2 ($\alpha$-phase) in coarse- and ultrafine-grained conditions was investigated. The obtained experimental results can be summarized as follows:

- Neutron irradiation significantly increases the strength and decreases the ductility of coarse-grained Ti Grade 2 as a result of the high density of defect clusters/dislocation loops created in the structure upon irradiation.
- Neutron irradiation has almost no influence on the mechanical performance of UFG material. No radiation-induced changes in microstructure were observed by TEM.
- UFG Ti is resistant to radiation damage in terms of ductility.
- Ultrafine-grained materials exhibit a high potential for applications in harsh radiation environments.
- Long-term thermal stability of the UFG microstructure under irradiation is also critical.

**Author Contributions:** Conceptualization, P.H.; methodology, M.Z.; validation, P.H.; investigation, P.Z., A.F. and M.Z.; data curation, T.K.; writing—original draft preparation, T.K. and P.Z.; writing—review and editing, M.J. and J.S.; visualization, T.K.; supervision, O.S.; project administration, O.S. All authors have read and agreed to the published version of the manuscript.

**Funding:** Financial support by the Czech Technological Agency TK01030153 is gratefully acknowledged. Partial financial support by CSF project 21-18652M is also gratefully acknowledged. T.K. acknowledges the financial support by the Integrated Infrastructure Operational Program under the project: Creation of a Digital Biobank to support the systemic public research infrastructure ITMS: 313011AFG4.

**Data Availability Statement:** Not applicable.

**Conflicts of Interest:** The authors declare no conflict of interest.

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
