# Peer review of "Influence of Neutron Irradiation on Microstructure and Mechanical Properties of Coarse- and Ultrafine-Grained Titanium Grade 2"

_metals, doi:10.3390/met12122180_

Round 1
Reviewer 1 Report
The authors report new interesting and useful results for specialists in radiation materials science.
I suggest accepting the manuscript for publication, but there are misprints in Figure 4: true strain (in Fig. 4c there is another misprint “srain”) is not measured in MPa. These misprints must be corrected.
Author Response
Reviewer 1
I suggest accepting the manuscript for publication, but there are misprints in Figure 4: true strain (in Fig. 4 c there is another misprint “srain”) is not measured in MPa. These misprints must be corrected.
The misprint in Fig. 4c was corrected and the wrong unit on x-axis was removed. The original Fig. 4 was replaced by the correct one.
Reviewer 2 Report
The paper by M. Janecek et al. reports an investigation about the mechanical properties of coarse and ultrafine grained conditions of Ti under neutron irradiation, to simulate some of the conditions found in nuclear reactors. While significant changes are found for the coarse-grained metal concerning yield stress and ductility, the ultrafine grained state is found much more resistant to radiation, both at room temperature and at 300 °C. The authors explain their results in terms of trapping of defects induced by radiation in a higher density of dislocations and grain boundaries.
I suggest a minor revision of the manuscript; here are my optional comments:
- line 73: dpa (I suppose 'displacement per atom'): acronyms should be explained the first time they appear; even if trivial to the authors, their explanation makes readable the paper also to a reader not familiar with the topic.
- Materials and methods: the number of specimens for each test should be specified, in order to evaluate the quality of the statistical analysis. In this connection, how did the authors estimate the uncertainties displayed in table 1? In other terms, are the error bars due to only to the scattering of the data or derive also from conservative estimates?
- Figure 4: maybe the differences of the curves could be better appreciated in figures a, b and d by reducing the maximum of the y-axis (e.g. 600 MPa for figure 4b).
- Typos: in figure 4c: true strain; line 131: detailed TEM…; line 168: ‘were’ observed
Author Response
Reviewer 2
- line 73: dpa (I suppose 'displacement per atom'): acronyms should be explained the first time they appear; even if trivial to the authors, their explanation makes readable the paper also to a reader not familiar with the topic.
The acronymus was explained.
- Materials and methods: the number of specimens for each test should be specified, in order to evaluate the quality of the statistical analysis. In this connection, how did the authors estimate the uncertainties displayed in table 1? In other terms, are the error bars due to only to the scattering of the data or derive also from conservative estimates?
In each condition we have executed 3 tensile tests and the scatter in table 1 corresponds to the scatter of data of individual curves. We have added this information in the Materials and Methods part.
- Figure 4: maybe the differences of the curves could be better appreciated in figures a, b and d by reducing the maximum of the y-axis (e.g. 600 MPa for figure 4b).
We have corrected all typos in Fig. 4 by replacing it for another one. However we wish to keep the scale in all Figs. 1-4 the same allowing the direct comparison of mechanical performace of CG and UFG material in non-irradiated and irradiated condition. By changing the scale of stress in y-axis one would loose this direct comparison.
- Typos: in figure 4c: true strain; line 131: detailed TEM…; line 168: ‘were’ observed
The complete Fig. 4 was changed, the typos were corrected as well as other comments of reviewer 1.
All other typos were also corrected.
Reviewer 3 Report
This is an interesting paper.
The reviewer only has one comment:
The authors claimed that "On the other hand, no remarkable changes were observed in the microstructure of UFG material upon irradiation. It can be attributed to the presence of a dense network of high-angle grain boundaries along with the high dislocation density stored in the original UFG material [15]". In order to improve the scientific soundnessm, the authors should provide more detailed evidence on this issue, instead just cite a reference.
Author Response
The authors claimed that "On the other hand, no remarkable changes were observed in the microstructure of UFG material upon irradiation. It can be attributed to the presence of a dense network of high-angle grain boundaries along with the high dislocation density stored in the original UFG material [15]". In order to improve the scientific soundnessm, the authors should provide more detailed evidence on this issue, instead just cite a reference.
It is well-known and accepted in the UFG community that severe plastic deformation (e.g. ECAP, HPT, ARB, etc.) introduces to the materials the high density of lattice defects – point defects and dislocations. The high strain introduced by SPD results in the rearrangement of dislocations and formation of subgrains and grains which is accompanied by the fragmentation of the original coarse grained microstructure and strong grain refinement (by a factor of 100-1000) as specified in the revised manuscript, see also the response to reviewer s 3, the first point). With increasing strain the original low angle grain boundaries are continuously transformed into high-angle grain boundaries whose surface area becomes huge. On the other hand, the irradiation by thermal neutrons results in the formation of point defects which are transformed into dislocation loops which are then absorbed by grain boundaries. Pl. see our response to rev. 3 question 2 and 3, for details and corresponding extension of our revised manuscript which reflects this fact in detail.
Additional text was added to the manuscript to provide the detailed evidence, one additional reference – a recent excellent review of the properties of UFG materials prepared by SPD techniques of ECAP and HPT – was also added.
Reviewer 4 Report
The effect of the neutron irradiation with dose level of 0.3 dpa on the microstructures and related tensile mechanical properties of Ti Grade 2 in coarse and ultrafine-grained condition was investigated in the present work. some problems should be explained and stated.
1. The grain size of CG and UFG Ti grade 2 in this work should be provided.
2. P6, line 1: Why did the grain boundaries can absorb the defects? If defects can be absorbed by the grain boundaries, when the dose level of neutron irradiation is more than 0.3 dpa, can grain boundaries absorb the all defects?
3. P6, line 11 and 12: The reason of defect clusters/dislocation loops induced by neutron should be provided.
Author Response
- The grain size of CG and UFG Ti grade 2 in this work should be provided.
Grain size of the coarse grained material as determined by EBSD is 12 ± 6 μm. The grain size in the UFG material is below the resolution of SEM. Therefore the ACOM-TEM technique was used and the avarage size was determined - 112 ± 104 nm. As expected the severe plastic deformation resulted in the strong microstructure refinement by the factor of 100.
The values of the grain size and the respective techniques of its determination were added to the text.
- P6, line 1: Why did the grain boundaries can absorb the defects? If defects can be absorbed by the grain boundaries, when the dose level of neutron irradiation is more than 0.3 dpa, can grain boundaries absorb the all defects?
Radiation-induced point defects (vacancies or interstitials) can be absorbed at grain boundaries because diffusion of these point defects along grain boundaries (pipe diffusion) is substantially faster than the bulk diffusion, see e.g. M.R. Sorensen, Y. Mishin, A.F. Voter, Phys. Rev. B, 62, 3658 (2000). Hence, vacancy or interstitial absorbed at grain boundary is quickly transported to the surface where it disappears. Because of this mechanism grain boundaries can in principle absorb all point defects created by radiation since defects absorbed at grain boundaries are not stored there but are quickly transported to the surface.
The text of the manuscript was extended to explain this effect in detail, also the reference was added.
- P6, line 11 and 12: The reason of defect clusters/dislocation loops induced by neutron should be provided.
Dislocation loops in neutron irradiated materials are formed by collapsing of large vacancy clusters, see e.g. S. S. Sheinin, Nature 194, 1272–1273 (1962). Energetic neutrons cause atom displacements and introduce Frenkel pairs (vacancies and interstitials). Vacancies in Ti at room temperature are mobile and agglomerate into vacancy clusters. The driving force for the agglomeration is the reduction of the surface energy. Large vacancy clusters collapse into dislocation loops, i.e. from a 3D defect into a 2D defect, since dislocation loops have lower energy. The typical development sequence of defects in neutron irradiated material is vacancies -> small vacancy clusters -> large vacancy clusters -> dislocation loops. Similarly interstitials agglomerate into clusters which develop into dislocation loops as well.
The text of the manuscript was extended to provide the explanation of the formation of defect clusters and/or dislocation loops, also the reference was added.
Round 2
Reviewer 3 Report
The authors have addressed my concerns. The paper can be accepted for publication.